# 2-Imidazoline Nitroxide Derivatives of Cymantrene

**DOI:** 10.3390/molecules27217545

**Published:** 2022-11-03

**Authors:** Kseniya Maryunina, Gleb Letyagin, Galina Romanenko, Artem Bogomyakov, Vitaly Morozov, Sergey Tumanov, Sergey Veber, Matvey Fedin, Evgeniya Saverina, Mikhail Syroeshkin, Mikhail Egorov, Victor Ovcharenko

**Affiliations:** 1International Tomography Center SB RAS, Institutskaya Str. 3a, 630090 Novosibirsk, Russia; 2Novosibirsk State University, Pirogova Str. 1, 630090 Novosibirsk, Russia; 3N. D. Zelinsky Institute of Organic Chemistry RAS, Leninsky Prospect, 47, 119991 Moscow, Russia

**Keywords:** manganeseorganic, cymantrene, nitroxide, paramagnets

## Abstract

The 2-imidazoline nitroxide derivatives of cymantrene—2-(*η*^5^-cyclopentadienyl)tricarbonylmanganese(I)-4,4,5,5-tetramethyl-4,5-dihydro-1*H*-imidazole-3-oxide-1-oxyl (NNMn) and 2-(*η*^5^-cyclopentadienyl)tricarbonylmanganese(I)-4,4,5,5-tetramethyl-4,5-dihydro-1*H*-imidazole-1-oxyl (INMn) were synthesized. It was shown that NNMn and INMn exhibit a sufficiently high kinetic stability both in solids and in solutions under normal conditions. Their structural characteristics, magnetic properties and electrochemical behavior are close to Re(I) analogs. This opens the prospect of using paramagnetic cymantrenes as prototypes in the design of Re(I) half-sandwiched derivatives for theranostics, where therapy is combined with diagnostics by magnetic resonance imaging due to the contrast properties of nitroxide radicals.

## 1. Introduction

The design of organometallic derivatives of nitroxide radicals [1,2,3] has a high potential for the development of new organic synthesis principles and the study of the mechanisms of catalytic reactions [4,5,6,7,8,9,10,11,12,13,14,15,16,17,18,19,20,21,22,23,24,25] as well as for the creation of polyfunctional magnetoactive materials exhibiting diverse intra- and intermolecular interactions and spin-exchange coupling [9,10,11,12,13,14,15,16,17,18,19,20,21,22,23,24,25,26,27,28,29,30,31,32,33,34,35,36,37,38,39,40,41], redox properties [34,35,36,37,38,39,40], and specific biological activity [1,2,3,39]. A number of studies have shown that organometallic nitroxides based on 2-imidazoline nitronyl nitroxides containing an M–C bond in the second position of the heterocycle [11,12,13,14,15,16,17,18,19,20,21,22,23] and sandwich/half-sandwich compounds bearing various nitroxide substituents in the cyclopentadienyl ring proved to be the most stable (Figure 1) [28,29,30,31,32,33,34,35,36,37,38,39,40,41].

High efficiency of Pd-catalyzed cross-coupling reactions of organogold carbene-like derivatives of 2-imidazoline nitronyl nitroxides with aryl- and vinyl-halides has been demonstrated as a reliable methodological tool for the obtaining of nitroxide types unavailable by other synthetic routes [12,13,14,15,16,17]. It has been shown that the introduction of various nitroxide substituents into ferrocene can also serve as a fruitful approach by which peculiar types of molecular magnets [28,29,30,31,32,33,34,35,36,37,38,39,40,41], spin labels with tunable spacing between different types of paramagnetic centers [36,37], miniature fast-charging electrochemical elements [38,39], and paramagnets with antioxidant activity have been obtained [40].

We would like to pay attention to the recently obtained first nitroxide-substituted derivative of cyrhetrene [41], that actually opened the opportunity of developing theranostics medicines on their basis. The therapeutic effect of radiological treatment with ^186/188^Re [42,43] isotopes in such compounds can be combined with monitoring and diagnosis by magnetic resonance imaging (MRI), where a paramagnetic organic fragment acts as a contrast agent [44,45]. In this regard, the development of methods for the synthesis and study of the corresponding nitroxide-substituted cymantrene derivatives becomes of essential importance. Paramagnetic cymantrenes can also be very promising due to their photodynamic therapeutic activity [46,47]; on the other hand, cymantrene analogs are more commercially available, and therefore it is more reasonable to use them to work out some synthetic steps when developing the design of target Re(I) or Tc(I) compounds [42,43]. This prompted us to synthesize 2-imidazoline nitroxide derivatives of cymantrene (Figure 2) and study their structure, magnetic properties, and electrochemical behavior.

## 2. Results and Discussion

Nitroxide-substituted cymantrenes were obtained by synthetic procedures similar to those for the previously studied cyrhetrene analogs [41]. The possibility of using commercially available cymantrene as a starting reagent on a larger scale allowed us to increase the yield of the target paramagnetic half-sandwiched nitroxides by ~1.5–2 times compared to cyrhetrene derivatives. Except for the need to protect NNMn and INMn from visible and UV light, they are stable in both solids and solutions and do not require an inert atmosphere, decreasing temperature or other special precautions for storage and handling in common laboratory conditions.

According to the data of SC XRD study, the bond lengths in the 2-imidazoline substituents and in the {Mn(C_Cp_)_5_(C_CO_)_3_} coordination units of NNMn and INMn are in the intervals typical of nitroxides and cymantrenes: N-O—1.24–1.28 Å, Mn-C_Cp_—2.12–2.15 Å, Mn-C_CO_—1.75–1.81 Å (Appendix A) [48]. In the crystal structure of NNMn/INMn there are two crystallographically independent molecules—**A** and **B** (Figure 1, Appendix A); in the INMn type **B** molecule, the NO group is disordered in two positions with a weight of 0.8/0.2. In type **A** molecules, the angle between the nitroxide fragment {O^●^-N-C = N(→O)} and the Cp cycle is 5.3° for NNMn and 6.0° for INMn, i.e., their planes are almost coplanar, whereas in type **B** molecules the angle between cycles is somewhat larger—19.8 and 13.8° for NNMn and INMn, respectively.

Mutual arrangement of NNMn and INMn molecules in crystalline structures is similar (Figure 1, Appendix A). Neighboring molecules **A** and **B** are combined into {**AB**} dimers with short distances between C atoms of cyclopentadienyl rings (NNMn: C_Cp_···C_Cp_ 3.29 Å; INMn: C_Cp_···C_Cp_ 3.32 Å). For nitronyl nitroxide NNMn this orientation relative to each other of molecules **A** and **B** is favorable for the formation of close contacts between NO groups (N···O_NO_ 3.96/4.01 and O_NO_···O_NO_ 4.13 Å), whereas for imino nitroxide INMn the nearest distances are realized between N_im_ atoms of adjacent molecules (N_im_···N_im_ 4.30 Å). Type **A** molecules for both NNMn and INMn are linked together in chains by H-bonds C_Cp_–H···O_NO_ (NNMn: C···O 3.24, H···O 2.62 Å; INMn: C···O 3.30, H···O 2.58 Å).

SQUID magnetometry measurements showed that NNMn and INMn (Figure 2, Appendix A) are paramagnets: *µ*_eff_ magnitude at 50–300 K is close to the theoretical spin-only value of 1.73 µ_B_ for the monoradical; the monotonic decrease in *µ*_eff_ with further lowering temperature from 30 to 2 K indicates the presence of antiferromagnetic exchange interactions. For NNMn nitronyl nitroxide, the optimal parameters of magnetic exchange interactions obtained by fitting the experimental dependence *µ*_eff_(T) within the model of exchange-coupled dimers (*H* = −2*JS*_R1_*S*_R2_, *S*_R1_ = *S*_R2_ = ½) [49] consist of ***J***^NNMn^ = −12.1 cm^−1^ and **g**^NNMn^ = 2.01. These results agree with the data of periodic quantum-chemical calculations and are confirmed by an independent molecular DFT calculation (***J***^QE^ mark—Quantum Espresso 6.2 package [50] and ***J***^ORCA^ mark—ORCA 5.0 quantum chemistry package [51]; Appendix A). The calculated values of intermolecular magnetic exchange interactions occurring in {**AB**} dimers with closely arranged NO groups (O_NO_···O_NO_ 4.13 Å) are ***J***^QE-NNMn^_NO···ON_ = −3.0 cm^−1^ and ***J***^ORCA-NNMn^_NO···ON_ = −3.15 cm^−1^. At the same time, the efficiency of the chain-wise magnetic exchange channel formed upon the H-bond of O_NO_ atom with the cyclopentadienyl ring (C···O = 3.24 Å) is much weaker: ***J***^QE-**NNMn**^_Cp···ON_ ~0 cm^−1^ and ***J***^ORCA-NNMn^_Cp···ON_ = −0.01 cm^−1^. The fitting of the *µ*_eff_(T) curve for INMn using the same dimer model yielded a much lower value of the magnetic exchange interactions—***J***^INMn^ = −0.8 cm^−1^ (**g**^INMn^ = 2.00). Unfortunately, in this case we failed to reproduce the weak antiferromagnetic exchange-coupling by either the molecular DFT method or the periodic DFT method used above. The much weaker magnetic exchange interactions for INMn as compared with NNMn is consistent with the larger distances between the {O^●^-N-C = N(→O)} fragments, where almost the entire spin density is concentrated (e.g., the closest N···N distances are 4.30 Å in INMn versus 4.08 Å in NNMn; Appendix A).

The EPR spectra of dilute solutions of NNMn and INMn (Figure 2) show characteristic five- and seven-line patterns for aromatically-substituted 2-imidazoline nitronyl and imino nitroxides, respectively [1,2,3]. Modeling parameters of EPR spectra used spin Hamiltonian in the form $\hat{H}=g_{iso}\mu_{B}HS_{z}+A_{iso}^{N1}SI^{N1}+A_{iso}^{N2}SI^{N2}$ (*S* = 1/2, $I^{N1}=I^{N2}$=1) [52] gave g*_iso_* = 2.009 for both compounds and hyperfine coupling constants $A_{iso}^{N1}$=$A_{iso}^{N2}$ = 0.721 mT for NNMn and $A_{iso}^{N1}$ = 0.915 mT, $A_{iso}^{N2}$ = 0.418 mT for INMn.

According to cyclic voltammetry (CV) data, for nitroxide-substituted cymantrenes, electron transfer appears only on the nitroxide fragments {NN}/{IN}, while the cymantrenous part {(Cp)M(CO)_3_} of the molecules remains unchanged in a wide range of electrochemical potentials (Figure 3, Appendix A). NNMn and INMn exhibit the irreversible reduction process characteristic of nitroxide radicals [53] when the potentials reach −1096 and −982 mV, respectively. The oxidation of nitronyl nitroxide NNMn at 793 mV is a one-electron completely reversible process, while the oxidation of imino nitroxide INMn occurs at higher potentials at 1284 mV and is irreversible. The distinction in the oxidation and reduction potentials for NNMn and INMn is consistent with the difference in the calculated energies of their *α*-SOMO and *β*-LUMO orbitals (ORCA 5.0 quantum chemistry package [51] with range-separated LC-BLYP functional and def2-TZVP basis set; Appendix A). The higher calculated magnitude of the orbital energy *α*-SOMO of −8.068 eV for NNMn than −8.560 eV for INMn agrees well with the lower value of its oxidation potential. The higher reduction potential for NNMn indicates its lower electron affinity, which corresponds to a higher calculated magnitude of the *β*-LUMO orbital energy: −0.466 eV for NNMn and −0.644 eV for INMn.

A comparison of the structure, magnetic, and electrochemical properties for the obtained nitroxide-substituted cymatrenes with the previously reported cyrhetrene [41] derivatives showed that they are complete analogs. The crystal structures of NNMn and INMn are isostuctural to one of the polymorphic modifications of the nitronyl nitroxide cyrhetrene NNRe-III (Appendix A). The similarity of the network of intermolecular contacts for NNMn and NNRe-III results in almost complete coincidence of the dependences *µ*_eff_(T) for them, as well as the parameters of intermolecular magnetic exchange interactions calculated by quantum chemistry [50] and fitted from the experimental data [49] (***J***^QE-NNMn^_NO···ON_ = −3.0 cm ^−1^, ***J***^NNMn^ = −12.1 cm ^−1^ and **g**^NNMn^ = 2.01; ***J***^QE-NNRe^_NO···ON_ = −5.3 cm^−1^, ***J***^NNRe-III^ = −12.3 cm^−1^ and **g**^NNRe-III^ = 2.03; Appendix A). Substitution Re(I) by Mn(I) in {(Cp)M(CO)_3_} moiety according to the results of quantum chemical calculations [51] and EPR spectroscopy data did not affect the character of the spin density spread in {O^●^-N-C = N→O}/{O^●^-N-C = N} fragments for the corresponding nitronyl and imino nitroxide derivatives of cymantrene and cyrhetrene (Appendix A). The calculated values of the orbital energies *α*-SOMO and *β*-LUMO for the cymantrene and cyrhetrene nitroxides are also close to each other, which agrees with the similar magnitudes of their electrochemical potentials from CV measurements (Figure 3, Appendix A).

In summary, the first persistent manganeseorganic nitroxide-substituted compounds NNMn and INMn were synthesized and characterized. They are kinetically stable in both solids and solutions and do not require any special precautions for their storage and handling. The physicochemical properties of NNMn and INMn are similar to those of the cyrhetrene analogues NNRe and INRe [41]. Taking into account that cymantrene derivatives are more commercially available, some synthetic steps can initially be developed on their basis and then adapted to produce the target Re(I) or Tc(I) compounds. Functionalized derivatives of both NNMn/INMn and NNRe/INRe could potentially be used to create new medicines where therapy with cymantrene or cyrhetrene-based moieties is combined with MRI diagnosis due to the contrast properties of nitroxide radicals.

## 3. Materials and Methods

### 3.1. General Procedures

The 2,3-bis(hydroxyamino)-2,3-dimethylbutane **BHA** [54], (*η*^5^-formylcyclopentadienyl)tricarbonylmanganese(I) [**(CHOCp)Mn(CO)_3_**] [55,56] were synthesized by the known procedures. The commercially available reagents and solvents for synthesis under Ar, electrochemical measurements and EPR study were purified, dried and degassed following standard literature methods [57] and/or using an MBRAUN MB SPS-800 system. The synthesis, purification and storage of cymantrene derivatives were carried out with darkening due to their light sensitivity. The reactions were monitored by TLC using «Alugram SIL G/UV254» and «POLYGRAM ALOX N/UV254» (“Macherey-Nagel”) sheets. Column chromatography was carried out with the use of SiO_2_ 0.04–0.0063 mm/230–400 mersh ASTM for column chromatography (“Macherey-Nagel”) and Al_2_O_3_ of chromatographic grade purchased from the Donetsk Plant of Chemical Reagents. The IR spectra of the samples pelletized with KBr were recorded on a «VECTOR-22» (Bruker, Karlsruhe, Germany). The melting points were determined on a melting point apparatus «Stuart» (SMP3). The microanalyses were performed on a «EURO EA3000» CHNS analyzer (HEKAtech, Webberg, Germany) at the Chemical Analytical Center of the Novosibirsk Institute of Organic Chemistry SB RAS.

### 3.2. Synthesis of Spin-Labeled Cymantrenes (Figure 3)

*2-(η^5^-cyclopentadienyl)tricarbonylmanganese(I)-4,4,5,5-tetramethyl-4,5-dihydro-1H-imidazole-3-oxide-1-oxyl* (**NNMn**). The mixture of freshly obtained [**(CHOCp)Mn(CO)_3_**] (0.580 g; 2.50 mmol) and **BHA** (0.555 g; 3.75 mmol) in dry EtOH (10 mL) was kept under Ar at room temperature for 72 h. Then, EtOH was evaporated, and the colorless residue was purified by column chromatography (Al_2_O_3_ 1.5 × 15 cm, Et_2_O as an eluent). After evaporation it was dried in a vacuum to obtain 0.760 g whitish powder of 2-(*η*^5^-cyclopentadienyl)tricarbonylmanganese(I)-4,4,5,5-tetramethylimidazolidine-1,3-diole (**BHAMn**). MnO_2_ (1.00 g) was added to a solution of **BHAMn** (0.760 g) in toluene (~15 mL) and the mixture was stirred for ~40 min at the 20 °C water bath. Then, the resultant dark-blue solution was filtered, the filtrate was evaporated, and the residue was purified by column chromatography (Al_2_O_3_ 1.5 × 15 cm, Et_2_O as an eluent). Gradual concentration of an Et_2_O:*n*-C_6_H_14_ = 1:5 solution by solvents evaporation from open flask at 15 °C gave aggregates of dark-blue elongated prismatic crystals. Yield: 0.570 g (75% per [**(CHOCp)Mn(CO)_3_**)]. *R*_f_ ~0.50 (Et_2_O) on the Aluminium oxide N/UV254 plates. **NNMn** is soluble in aromatic hydrocarbons, halogen-substituted hydrocarbons, acetone, Et_2_O and alcohols, moderately soluble in saturated hydrocarbons, and insoluble in water. IR spectrum (KBr) ν: 2994, 2023, 1934, 1560, 1453, 1429, 1398, 1372, 1143, 1037, 869, 631, 542 cm^−1^. Mp 139–141 °C (decomp). Found (%): C, 50.22; H, 4.83; N, 7.75. Calculated for C_15_H_16_MnN_2_O_5_ (%): C, 50.15; H, 4.49; N, 7.80.*2-(η^5^-cyclopentadienyl)tricarbonylmanganese(I)-4,4,5,5-tetramethyl-4,5-dihydro-1H-imidazole-1-oxyl* (**INMn**). Oxidation of the condensation product of [**(CHOCp)Mn(CO)_3_]** (0.235 g; 1.01 mmol) and **BHA** (0.300 g; 2.03 mmol) omitting its purification by column chromatography resulted in **INMn** as a main product and **NNMn** as an admixture. **INMn** was separated by column chromatography (Al_2_O_3_ 1.5 × 15 cm, Et_2_O as an eluent) and recrystallized from an Et_2_O:*n*-C_6_H_14_ mixture. Yield: 0.073 g (25% per [**(CHOCp)Mn(CO)_3_**]), orange elongated plates. **INMn** soluble in aromatic hydrocarbons, halogen-substituted hydrocarbons, saturated hydrocarbons, acetone, Et_2_O, alcohols, insoluble in water. *R*_f_ ~0.95 (Et_2_O) on the Aluminium oxide N/UV254 plates. Mp 89–92 °C (decomp). IR spectrum (KBr) ν: 2982, 2023, 1933, 1587, 1437, 1413, 1377, 1165, 1038, 843, 633, 542 cm^−1^. Found (%): C, 52.47; H, 4.65; N, 8.11. Calculated for C_15_H_16_MnN_2_O_5_ (%): C, 52.49; H, 4.70; N, 8.16.

**Scheme 3 molecules-27-07545-sch003:**
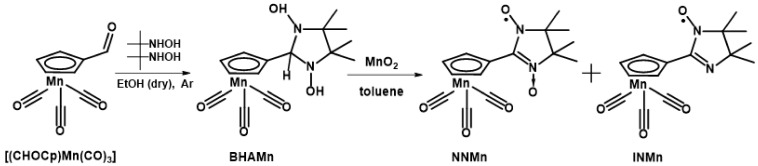
Synthesis of NNMn and INMn.

### 3.3. Single Crystals X-ray Crystallography

The intensity data for single crystals were collected on Bruker AXS diffractometers—a SMART APEX II (Mo Kα radiation) and an APEX DUO (Cu Kα radiation) with a Cobra low-temperature accessory (Oxford Cryosystem). The structures were solved by direct methods and refined by full-matrix least-squares in an anisotropic approximation for all non-hydrogen atoms. The H atoms were calculated geometrically and refined in a riding model. All calculations on structure solution and refinement were performed with SHELXL (2016/4 and 2018/3) software. Crystallographic data, selected bond lengths, angles and intermolecular distances can be found in Appendix A. Crystallographic data have been deposited with the Cambridge Crystallographic Data Centre, deposition numbers 2,182,686 (NNMn) and 2,182,685 (INMn). These data can be obtained free of charge via http://www.ccdc.cam.ac.uk/conts/retrieving.html, accessed on 27 October 2022 (or from the CCDC, 12 Union Road, Cambridge CB2 1EZ, UK; Fax: +44-1223-336033; E-mail: deposit@ccdc.cam.ac.uk).

### 3.4. Variable-Temperature SQUID Magnetometry

Magnetic measurements were carried out on MPMSXL SQUID magnetometer (Quantum Design) in the temperature range 2−300 K in a magnetic field of up to 5 kOe. The paramagnetic components of the magnetic susceptibility χ were determined with allowance for the diamagnetic contribution evaluated from the Pascal constants [58]. The effective magnetic moment was calculated as *μ*_eff_ = ((3k/N_A_μ_B_^2^)*χ*T)^1/2^ ≈ (8χT)^1/2^.

### 3.5. Quantum-Chemical Calculations

The isotropic exchange parameters (Appendix A) were estimated within the broken-symmetry approach of Yamaguchi and co-workers [59]. All periodic DFT + *U* calculations were performed using the crystallographically determined geometries and an approach based on calculations of similar systems [60] utilizing pseudo-potential PW-SCF code of Quantum Espresso 6.2 package [50]. We used the nonlinear core-corrected ultrasoft pseudo-potentials of type X.pbe-van_ak.UPF (X—is a symbol of chemical element) with the PBE exchange-correlation functional. The kinetic energy cutoffs for wave functions and charge density are 50 and 400 Ry, respectively. The integration in the *k* space was performed over the mesh 2 × 2 × 2 in the first Brillouin zone as in Monkhorst−Pack scheme [61] with a displacement of *k*-grid at the center of the Brillouin zone and the Gaussian smoothing of 0.136 eV. The Hubbard correlations on Mn and O sites were taken into account within the framework of the Dudarev version of GGA + *U* approach [62] with the values *U*_d_(Mn) = 5.0 eV [63], *U*_p_(O) = 5.0 eV [64]. The results of periodic DFT + *U* calculations are confirmed by independent molecular DFT calculations that were performed by ORCA 5.0 quantum chemistry package [51] using the TPSSh hybrid functional in ma-def2-TZVP basis set, augmented by s-, p-diffuse functions, which is important for correct calculation of distance interactions.

The total Mulliken spin density values localized on atoms of {O^●^-N-C = N→O} or {O^●^-N-C = N} fragments and metal centers and energy values of *α*-SOMO and *β*-LUMO orbitals for NNM and INM (M = Mn, Re) were estimated by molecular DFT calculations with optimized geometries of the complexes, carried out with ORCA 5.0 quantum chemistry package [51] using range-separated LC-BLYP functional and def2-TZVP basis set (Appendix A).

### 3.6. EPR Spectroscopy

Continuous wave (CW) X-band (9.88 GHz) EPR measurements were carried out on a Bruker EMX spectrometer at room temperature. Spectra were recorded at a microwave power of ~2 mW and a modulation amplitude of 0.01 mT at 100 kHz. Diluted toluene solutions of NNMn and INMn (~10^−3^ mM) were studied; samples were preliminarily degassed by freeze–pump–thaw cycles. EPR data were modeled with the EasySpin toolbox [52].

### 3.7. Cyclic Voltammetry

Cyclic voltammetry (CV) measurements were performed with a PC-piloted digital potentiostat IPC-Pro-MF (Econix). The experiments were carried out in a 10-mL five-neck glass conic electrochemical cell equipped with a water jacket for thermostating. As a working electrode, a glass carbon (GC) disk (*d* = 1.7 mm) was used, polished before each run; a platinum wire was used as an auxiliary electrode. The potentials are referred to the AgCl/KCl_sat_ electrode separated from the analyte by an electrolytic bridge filled with supporting electrolyte (0.1 M Bu_4_NBF_4_/MeCN). Solution deaeration was carried out by purging them with highly pure Ar before recording every CV curve and Ar passing over it for the duration of the experiments. The approximation to zero current of the potentials of cathodic and anodic peaks to the determination of half-wave potentials *E*^0′^ was carried out as described elsewhere [53,65].

## Data Availability

Not applicable.

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
