# Peer review of "2-Imidazoline Nitroxide Derivatives of Cymantrene"

_molecules, 2022, doi:10.3390/molecules27217545_

Round 1
Reviewer 1 Report
Two types of nitroxide radical were attached to the cyclopentadiene ring of the Mn(CO)3cp compound called cymantrene. The authors previously published Re(I) analogs.
Intermolecular spin interactions were estimated from magnetochemistry of the concentrated solids. The magnitude is close to the spin-only value, with small antiferromagnetic interactions revealed at low temperature. CW EPR were measured using 1 mM solutions from which O2 was removed. Cyclic voltammetry measurements showed that redox occurs only on the nitroxide moiety. Quantum chemistry calculations were made.
The results are clearly presented and data in the tables in the supplementary information support the interpretation. There is appropriate warning about the sensitivity of the compounds to light.
Author Response
Two types of nitroxide radical were attached to the cyclopentadiene ring of the Mn(CO)3cp compound called cymantrene. The authors previously published Re(I) analogs.
Intermolecular spin interactions were estimated from magnetochemistry of the concentrated solids. The magnitude is close to the spin-only value, with small antiferromagnetic interactions revealed at low temperature. CW EPR were measured using 1 mM solutions from which O2 was removed. Cyclic voltammetry measurements showed that redox occurs only on the nitroxide moiety. Quantum chemistry calculations were made.
The results are clearly presented and data in the tables in the supplementary information support the interpretation. There is appropriate warning about the sensitivity of the compounds to light.
Response: We thank Reviewer for thoughtful comments and appreciate for the comprehensive evaluation of our manuscript.
Reviewer 2 Report
This article entitled ‘Nitroxide-substituted cymantrene’ is an interesting contribution to the chemistry of nitroxide cymantrene species and their possible applications for therapy. It may deserve to be published after corrections and comments have been considered by the authors.
The title of the article seems too general and it should be modified for better highlighting the reported results.
The abstract has to be rewritten. It does not allow to know shortly, before reading, the results obtained and to understand the goal of the work. For example, what is the meaning and the reason to qualify of ‘persitent’ the studied complexes. It is indicated that the complexes are of ‘sufficiently high kinetic stability’ ; that is not understandable if not explicitly explained the reasons why the authors are looking for sufficient stability of such species.
A carefully reading by the authors of their manuscript will allow them to correct few typos (replace ‘2nd ‘by ‘second’ l19, for example) and to improve the english writing of some sentences .
Author Response
This article entitled ‘Nitroxide-substituted cymantrene’ is an interesting contribution to the chemistry of nitroxide cymantrene species and their possible applications for therapy. It may deserve to be published after corrections and comments have been considered by the authors.
Response: We thank Reviewer for a comprehensive evaluation of our manuscript and for valuable comments and suggestions. All changes and insertions made to the online submitted version of the revised manuscript were marked up in red using the “Track Changes” function.
The title of the article seems too general and it should be modified for better highlighting the reported results.
Response: As recommended by Reviewer, we change the title of manuscript from “Nitroxide-substituted cymantrene” to more detailed one – “2-Imidazoline nitroxide derivatives of cymantrene” for better highlighting our results.
The abstract has to be rewritten. It does not allow to know shortly, before reading, the results obtained and to understand the goal of the work. For example, what is the meaning and the reason to qualify of ‘persitent’ the studied complexes. It is indicated that the complexes are of ‘sufficiently high kinetic stability’ ; that is not understandable if not explicitly explained the reasons why the authors are looking for sufficient stability of such species.
Response: We agree with Reviewer and rewrote the abstract according to his suggestion. The term “stable nitroxide radical” means “…can be obtained in pure form, stored and handled in the laboratory with no more precautions than that normally observed when working with conventional organic compounds [Chem. Rev., 1978, 78, 37]”. The long living ability of nitroxides is most efficiently provided by the kinetic factors, e. g. steric hindrance of the nitroxide group at the introduction of bulky substituents into the molecular structure [L. B. Volodarsky, V. A. Reznikov, V. I. Ovcharenko, Synthetic Chemistry of Stable Nitroxides, Boca Raton, FL: CRC Press, 1994]. This feature is emphasized in the expression “high kinetic stability”, which is widely used in various versions in the literature. In this case, the expression “high kinetic stability” means that the obtained 2-imidazoline nitroxide derivatives of cymantrene are long-lived radicals. The fact these compounds can be stored in the form of solids and solutions for a long time at room temperature and in a non-inert atmosphere already indicates high potential for their various application. The functionalization of nitroxide-substituted cymantrenes in order to make them more suitable for use in pharmacology is under way.
Text editing: Abstract: The 2-imidazoline nitroxides derivatives of cymantrene – 2-(η5-cyclopentadienyl)tricarbonylmanganese(I)-4,4,5,5-tetramethyl-4,5-dihydro-1H-imidazole-3-oxide-1-oxyl (NNMn) and 2-(η5-cyclopentadienyl)tricarbonylmanganese(I)-4,4,5,5-tetramethyl-4,5-dihydro-1H-imidazole-1-oxyl (INMn), were synthesized. It was shown that NNMn and INMn exhibit a sufficiently high kinetic stability both in solids and in solutions under normal conditions. Their structural characteristics, magnetic properties and electrochemical behavior are close to Re(I) analogs. These open the prospect of use paramagnetic cymantrenes as prototypes in the design of Re(I) half-sandwiched derivatives for theranostics, where therapy is combined with diagnostics by magnetic resonance imaging due to the contrast properties of nitroxide radicals.
A carefully reading by the authors of their manuscript will allow them to correct few typos (replace ‘2nd ‘by ‘second’ l19, for example) and to improve the english writing of some sentences.
Response: Thank you for carefully reading the text of our manuscript. We made this correction and edited the other article text of checking the English grammar and misprints.

Reviewer 3 Report
The manuscript by Dr. Ovcharenko et al. reported synthesis, crystal structure and magnetic and electrochemical properties of nitroxide-substituted cymantrene. The nitroxides are potentially interesting as a functional compounds, and the experimental were performed adequately. This manuscript has merit to be published in Molecule.
Some correction and revisions are requested as fillows:
(1) Page 2, line 40: could be ‘between’ ...
(2) Line 70: 2.12–2.15 (?)
(3) Line 80: The pi-stacking is not obvious in Fig. 1. Are there any pi-stacking interaction? At least I could not find them using CIF files.
(4) Page 3, line 97: Why the experimental (fitted) and calculated magnetic exchange parameter, J, for NNMn are so different? Is the exchange-coupled dimer model reasonable for these crystals?
(5) Page 4, line 139: What does isotypical means? Isn’t it isomorphous or isostructural?
(6) Page 5, line 191: it should be ‘...tricarbonylmanganese(I)-...‘.
(7) Page 6, line 193: What is the temperature of the water bath? It would be better to write 40 min (than 0.66 h).
(8) Line 219: 0.073 g (?)
Author Response
The manuscript by Dr. Ovcharenko et al. reported synthesis, crystal structure and magnetic and electrochemical properties of nitroxide-substituted cymantrene. The nitroxides are potentially interesting as a functional compounds, and the experimental were performed adequately. This manuscript has merit to be published in Molecule.
Response: We thank Reviewer for a comprehensive evaluation of our manuscript and for valuable comments and suggestions.
Some correction and revisions are requested as follows:
(3) Line 80: The pi-stacking is not obvious in Fig. 1. Are there any pi-stacking interaction? At least I could not find them using CIF files.
Response: We have reconsidered SC XRD data on the overlap of the π-systems of cyclopentadienyl rings and agree with Reviewer that it is not significant enough to efficient π-stacking. Corresponding changes were made to the text of the manuscript.
Text editing: “Neighboring molecules A and B are combined into {AB} dimers with short distances between C atoms of cyclopentadienyl rings (NNMn: CCp...CCp 3.29 Å; INMn: CCp...CCp 3.32 Å).” instead of “The {AB} dimers are formed due to π-stacking of the cyclopentadienyl rings of molecules A and B(NNMn: CCp...CCp 3.29 Å, ÐCp…Cp 20.0°; INMn: CCp...CCp 3.32 Å, ÐCp…Cp 23.1°).”.
(4) Page 3, line 97: Why the experimental (fitted) and calculated magnetic exchange parameter, J, for NNMn are so different? Is the exchange-coupled dimer model reasonable for these crystals?
Response: When considering a network of intermolecular contacts, of particular interest is the environment of N–O groups, since almost all of the spin density is concentrated on their π-antibonding orbitals. Coplanar arrangement of the {ONCN(O)} nitroxide fragment and the Cp ring in nitroxides under study are could be favorable for the spread of the spin density to the conjugated aromatic {ONCN(O)–Cp} system. The performed periodic quantum-chemical calculations made it possible to establish that the strongest magnetic exchange interactions for NNMn are realized at the close location of NO groups of neighboring molecules arranged in pairs. At the same time, it was shown that the efficiency of the magnetic exchange channel formed upon the contact of NO groups with the cyclopentadienyl ring is much weaker. After receiving the revision, we made additional molecular DFT calculations (ORCA 5.0 TPSSh/ma-def2-TZVP) of the potentially most efficient magnetic exchange channels with obtaining similar values of the parameters of magnetic exchange interactions JORCANO…ON ~ -3.15 cm-1 and JORCACp…NO ~ -0.08 cm-1. Thus, the most efficient channels of magnetic exchange interactions were found in pairwise arranged molecules of NNMn nitronyl nitroxides, which makes it justified to use the dimer model for fitting the experimental data. Our numerous attempts to carry out quantum-chemical calculations by different approaches did not allow obtaining reliable estimation of magnetic exchange parameters for INMn imino nitroxide. E.g. according to the results of DFT calculation of ORCA TPSSh/ma-def2-TZVP gave JORCANO…NO = +1.8 cm-1 for the {AB} dimer and JORCACp…NO = +0.01 cm-1 for the chain-wise pathway. That is, according to the results of DFT calculations, these channels are characterized by weak ferromagnetic exchange-coupling, while the experimental dependence µeff(T) manifests antiferromagnetic exchange interactions fitted with Jdimer = -0.8 cm-1 optimal parameter. Appropriate explanations were included in the text of Section 2. Results and Discussion of the manuscript.
Calculations for the same magnetic exchange channel by several methods using various bases can give different values of the parameters of magnetic exchange interactions. A good agreement between the calculated values and the fitting of experimental data can both indicate in favor of the applicability of this calculation approach for such multi-spin system, or be pure chance. In our case, we did not aim to achieve complete agreement between the fitted and calculated parameters of the magnetic exchange, but only at a qualitative level to identify the most effective channels of magnetic exchange interaction, which was done. The difference between the values of the optimal parameters obtained in the course of fitting with the data of quantum chemical calculations is quite common, and in most cases there is an underestimation of the parameters of magnetic exchange interactions at their absolute values of about 10-20 cm-1. The following hypothesis can be assumed as the reason for such discrepancy: in most cases, DFT calculations are carried out for the structure obtained at room temperature, but as the temperature decreases, as a rule, the solid phase is compressed, which may be accompanied by some reduction in intermolecular distances and an enhancement of magnetic exchange interactions. Although the effect of thermal compression is insignificant, even a minor decreasing of contacts between fragments carrying spin density (~0.05 Å) can manifest itself in strengthening the magnetic exchange interactions (~1–5 cm-1). Verification of this assumption requires a systematic study of a specially selected group of objects by SC XRD at different temperatures with the DFT calculation of the parameters of magnetic exchange interactions and magnetochemical study in a wide temperature range, which is beyond the scope of our submitted study.
Text editing: For NNMn nitronyl nitroxide, the optimal parameters of magnetic exchange interactions obtained by fitting the experimental dependence µeff(T) within the model of exchange-coupled dimers (H = ‑2JSR1SR2, SR1 = SR2 = ½) [49] consist of JNNMn = ‑12.1 см‑1 and gNNMn = 2.01. These results agree with the data of periodic quantum-chemical calculations and confirmed by an independent molecular DFT calculation (JQE mark – Quantum Espresso 6.2 package [50] and JORCA mark – ORCA 5.0 quantum chemistry package [53]; Table S4). The calculated values of intermolecular magnetic exchange interactions occurring in {AB} dimers with closely arranged NO groups (ONO…ONO 4.13 Å) are JQE-NNMnNO…ON = ‑3.0 cm‑1 and JORCA‑NNMnNO…ON = ‑3.15 cm‑1. At the same time, the efficiency of the chain-wise magnetic exchange channel formed upon the H-bond of ONO atom with the cyclopentadienyl ring (C…O = 3.24 Å) is much weaker: JQE-NNMnCp…ON ~ 0 cm‑1 and JORCA-NNMnCp…ON = ‑0.01 cm‑1. The fitting of the µeff(T) curve for INMn using the same dimer model yielded a much lower value of the magnetic exchange interactions – JINMn = ‑0.8 cm‑1 (gINMn = 2.00). Unfortunately, in this case we failed to reproduce the weak antiferromagnetic exchange-coupling by neither the molecular DFT method nor the periodic DFT method used above.
We added information about the details of periodic DFT+U and molecular DFT calculations in Section 4.5.
Text editing: We used the nonlinear core corrected ultrasoft pseudo-potentials of type X.pbe-van_ak.UPF (X – is a symbol of chemical element) with the PBE exchange-correlation functional… … The results of periodic DFT+U calculations are confirmed by independent molecular DFT calculation that were performed by ORCA 5.0 quantum chemistry package [53] using the TPSSh hybrid functional in ma-def2-TZVP basis set, augmented by s-, p-diffuse functions, which is important for correct calculation of distance interactions.
(1) Page 2, line 40: could be ‘between’ ...
(2) Line 70: 2.12–2.15 (?)
(5) Page 4, line 139: What does isotypical means? Isn’t it isomorphous or isostructural?
(6) Page 5, line 191: it should be ‘...tricarbonylmanganese(I)-...‘.
(7) Page 6, line 193: What is the temperature of the water bath? It would be better to write 40 min (than 0.66 h).
(8) Line 219: 0.073 g (?)
Response to comments 1,2, 5-7: We thank the Reviewer for carefully reading our manuscript and we apologize for our mistakes and misprints. We edited the text of the article, corrected some phrases and sentences, checking the English grammar and typos.
